# Exploring Inflammasome Complex as a Therapeutic Approach in Inflammatory Diseases

Sharmim Sultana [1,†], Thanh Doan Viet [2,†], Tasmiha Amin [1], Esha Kazi [1], Luigina Micolucci [3], Abul Kalam Mohammad Moniruzzaman Mollah [1], Most Mauluda Akhtar [1] and Md Soriful Islam [2,*]

1 Science and Math Program, Asian University for Women, Chattogram 4000, Bangladesh; sharmim.sultana@post.auw.edu.bd (S.S.); tasmiha.amin@post.auw.edu.bd (T.A.); esha.kazi@post.auw.edu.bd (E.K.); moniruzzaman.mollah@auw.edu.bd (A.K.M.M.M.); mauluda82@gmail.com (M.M.A.)
2 Department of Gynecology and Obstetrics, Division of Reproductive Sciences & Women's Health Research, Johns Hopkins Medicine, Baltimore, MD 21205, USA; tdoan7@jhu.edu
3 Department of Clinical and Molecular Sciences, Università Politecnica Delle Marche, Via Tronto 10/A, 60126 Ancona, Italy; l.micolucci@univpm.it
* Correspondence: soriful84@gmail.com or mislam18@jhmi.edu; Tel.: +1-410-614-2000
† These authors contributed equally to this work.

**Abstract:** Inflammasomes, a group of multiprotein complexes, are essential in regulating inflammation and immune responses. Several inflammasomes, including nucleotide-binding domain leucine-rich repeat-containing protein 1 (NLRP1), NLRP3, NLRP6, NLRP7, NLRP12, interferon-inducible protein 16 (IFI16), NOD-like receptor family CARD domain-containing protein 4 (NLRC4), absent in melanoma 2 (AIM2), and pyrin, have been studied in various inflammatory diseases. Activating inflammasomes leads to the processing and production of proinflammatory cytokines, such as interleukin (IL)-1β and IL-18. The NLRP3 inflammasome is the most extensively studied and well characterized. Consequently, targeting inflammasomes (particularly NLRP3) with several compounds, including small molecule inhibitors and natural compounds, has been studied as a potential therapeutic strategy. This review provides a comprehensive overview of different inflammasomes and their roles in six inflammatory diseases, including multiple sclerosis, Alzheimer's disease, Parkinson's disease, atherosclerosis, type 2 diabetes, and obesity. We also discussed different strategies that target inflammasomes to develop effective therapeutics.

**Keywords:** inflammasome; inflammatory disease; NLRP3; multiple sclerosis; Alzheimer's disease; Parkinson's disease; atherosclerosis; type 2 diabetes; MCC950; tranilast





## 1. Introduction

The immune system machinery distinguishes between normal interactions ("self") and foreign threats ("non-self"). The presence of invading pathogens, toxins, or allergens initiates immune response, including innate and adaptive immunity [1]. Inflammation is an innate immune response induced by harmful stimuli; however, inadequate inflammation may lead to persistent infection, whereas uncontrolled inflammation can contribute to the development of chronic inflammatory diseases. The innate immune system relies on pattern recognition receptors (PRRs) that detect conserved molecular patterns on pathogens, known as pathogen-associated molecular patterns (PAMPs) and danger-associated molecular patterns (DAMPs) [2].

Inflammasomes are a group of multiprotein complexes that act as sensors of cellular stress and danger signals, activating inflammatory responses. They play an important role in the processing and secretion of proinflammatory cytokines, such as interleukin (IL)-1β and IL-18 [3]. Inflammasomes consist of a sensor protein, an adapter protein ASC (apoptosis-associated speck-like protein containing a caspase recruitment domain

(CARD)), and proinflammatory caspase-1 [4]. They play an essential role in maintaining body homeostasis and controlling the magnitude of inflammation in both normal physiological processes and pathological states [5]. Inflammasomes are associated with many autoimmune and autoinflammatory diseases, including neurodegenerative and metabolic disorders, which may result from dysregulated inflammasome activity [6]. Therefore, inhibitors of inflammasomes may contribute to the development of new therapeutics against inflammasome-related diseases [7].

This review provided an overview of the role of inflammasomes in inflammatory diseases and explored their therapeutic implications. Particularly, we discussed the role of various inflammasome complexes, including nucleotide-binding domain leucine-rich repeat-containing protein 1 (NLRP1), NLRP3, NLRP6, NLRP7, NLRP12, interferon-inducible protein 16 (IFI16), NOD-like receptor family CARD domain-containing protein 4 (NLRC4), and absent in melanoma 2 (AIM2) in multiple inflammatory conditions, including multiple sclerosis, Alzheimer's disease, Parkinson's disease, atherosclerosis, type 2 diabetes, and obesity. We also discussed therapeutic approaches targeting inflammasome complexes in these conditions.

## 2. Methods

This article presents a comprehensive overview of the existing literature regarding the significant role of inflammasomes in various inflammatory diseases and the developments in therapeutic approaches. An extensive search of the literature was conducted using electronic databases, such as PubMed and Google Scholar, until May 2023. The review articles were used as a source of additional articles related to inflammasomes and inflammatory diseases. We used a combination of the following keywords. These include "inflammasome", "inflammatory disease", "NLRP3", "NLRP1", "NLRC4", "NLRP6", "NLRP7", "NLRP12", "IFI16", "Pyrin" "IL-1β", "IL-18", "AIM2", "Non-canonical Inflammasomes", "multiple sclerosis", "experimental autoimmune encephalomyelitis", "Alzheimer's disease", "Parkinson's disease", "atherosclerosis", "type 2 diabetes", "obesity", "rheumatoid arthritis", "inflammatory bowel disease", "gout", "psoriasis", "systemic lupus erythematosus", "inflammasome therapeutic", "inflammasome inhibitor" "tranilast", "NLRP3 inhibitor", and "MCC950", "ketotifen", "IC100", "fenamates", and "canakinumab". For this article, we only considered articles published in English.

## 3. The Inflammasome: Mechanisms of Activation

Inflammasomes are multiprotein complexes composed of three main components: (1) a pattern recognition receptor (PRR), (2) an adaptor protein, and (3) an effector protein (Figure 1). The function of PRR is to detect danger signals. At the same time, the adaptor protein (carrying an ASC) promotes the assembly of the complex and establishes a connection with the effector protein (usually a caspase enzyme) [3,4] (Figure 1). Inflammasome assembly may occur due to a variety of triggers or stimuli. It can be initiated by detecting various danger signals, such as PAMPs and DAMPs, released through infection, tissue damage, or cellular stress [8]. Accumulating inflammasome complex leads to the activation of caspase-1 through proximity-induced self-cleavage. After caspase-1 activation, it induces the maturation of specific proinflammatory cytokines, such as IL-1β and IL-18, by proteolytically cleaving pro-IL-1β and pro-IL-18 (Figure 2). These cytokines are important in inflammation and immune response [3,5]. Activated caspase-1 can also cleave gasdermin D (GSDMD), which may induce a form of lytic programmed cell death called pyroptosis. The activation of this process warrants the detection of cytosolic contamination or perturbations and eliminates compromised cells, thus resulting in defense against intracellular infection and stimulation of the inflammatory response [9].

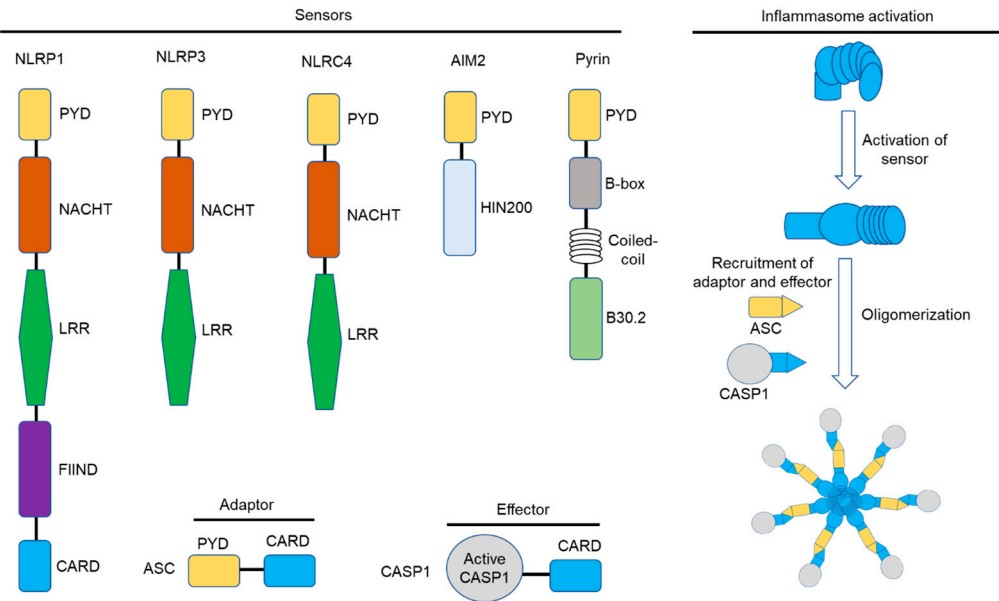

**Figure 1.** Domain organization and activation of the most known canonical inflammasomes. On the left, the NLRP1, NLRP3, NLRC4, AIM2, Pyrin, ASC, and CASP1 domains' organization. On the right, a schematic view of the canonical activation process. The typical inflammasome contains three components: sensors, adapters, and effector proteins. Sensors oligomerize upon activation and recruit adapter and effector proteins to the inflammasome complex. Modified from Christgen et al. [10] and Hamarsheh and Zeiser [11].

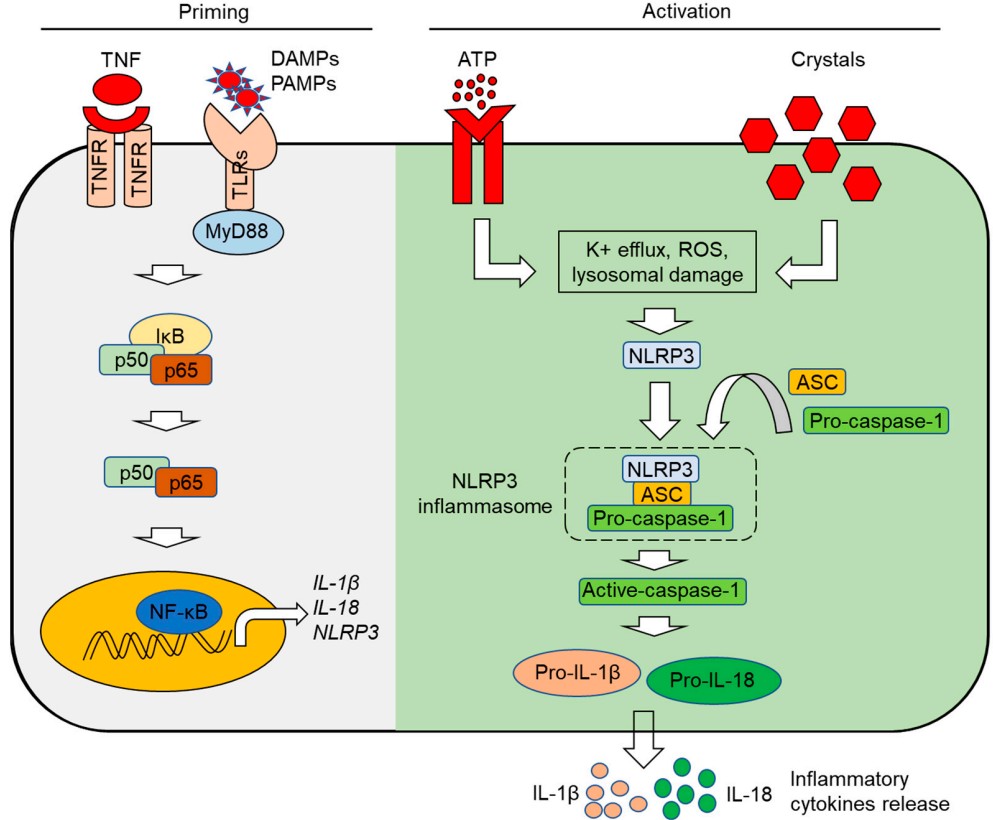

**Figure 2.** NLRP3 inflammasome activation. Modified from Christgen et al. [10] and Hamarsheh and Zeiser [11].

Several types of inflammasomes have been identified, such as the NLRP (NOD-like receptor family, pyrin domain-containing) inflammasomes, AIM2 inflammasome, and pyrin inflammasome [12]. These may exhibit both cytoplasmic and nuclear sensor molecules that can be activated by different stimuli. Multiple studies have demonstrated how inflammation is activated in specific types of tissues under different conditions [13,14]. One study found that increased IL-6 levels in pregnant mice result in cell-specific effects, specifically, immediate changes in fetal cells and long-term effects on downstream epithelial cell function [15]. However, in general, the mechanism for tissue selectivity of inflammasome activation is poorly understood due to a dearth of limited, direct studies. Understanding the tissue specificity of inflammasome activation is critical for developing novel targeted therapies.

### 3.1. The NLRP1 Inflammasome

NLRP1, also known as NALP1, NAC, DEFCAP, CLR17.1, and CARD7, was the first inflammasome extensively studied [16]. Typically, the NLRP1 inflammasome consists of NLRP1, the adaptor protein ASC, and the effector protein pro-caspase-1. ASC, which contains both a pyrin domain (PYD) and a CARD domain, may serve as a bridge between NLRP1 and pro-caspase-1 [17]. NLRP1 consists of three main domains: a C-terminal leucine-rich repeat (LRR) domain, a central nucleotide-binding and oligomerization (NACHT) domain, and an N-terminal PYD (Figure 1). The LRR domain recognizes and binds to specific ligands, while the NACHT domain facilitates the assembly of NLRP1 into a functional inflammasome complex [18]. Regarding the NLRP1 inflammasome, the N-terminal PYD is only found in humans and other non-rodent species [17]. Unlike other NLRs, NLRP1 proteins possess a different structural arrangement at the C-terminus. They feature a function to find domain (FIIND) consisting of ZU5 and UPA subdomains and undergo autoproteolysis between them [19]. The mouse genome encodes three NLRP1 paralogs, *NLRP1A*, *NLRP1B*, and *NLRP1C* [20], and NLRP1 has been highly variable among inbred rodent strains [20,21]. NLRP1 activation typically involves two steps. First, the LRR domain of NLRP1 detects and interacts with specific ligands, such as bacterial toxins or viral proteins [17]. This binding induces a conformational change in NLRP1, leading to the exposure of its PYD domain. Second, the exposed PYD domain of NLRP1 binds with the adaptor protein ASC, which then recruits and activates pro-caspase-1. Active caspase-1 cleaves IL-1β and IL-18 into mature forms, leading to inflammation and immune responses [22].

### 3.2. The NLRP3 Inflammasome

NLRP3 (NALP3), or cryopyrin, is an extensively studied inflammasome. The NLRP3 inflammasome comprises three main components: NLRP3 as the sensor molecule, ASC as the adaptor protein, and caspase-1 as the effector protein [23] (Figures 1 and 2). These three components form the NLRP3 inflammasome complex, which activates caspase-1 and proinflammatory cytokines' subsequent processing and secretion. The active regulation of the NLRP3 inflammasome involves a two-step mechanism (Figure 2). An initial non-activating stimulus, known as "priming", is required to trigger the expression of critical inflammasome components. This priming step can be triggered by various PRRs, such as TLR4 and NOD2, as well as cytokine receptors, such as TNFR and IL-1R [23]. Following activation, these receptors activate the nuclear factor "kappa-light-chain-enhancer" of the activated B cell (NF-κB) signaling pathway, resulting in the transcription and translation of NLRP3 as well as pro-IL-1β and pro-IL-18 (Figure 2). After the priming step, a secondary stimulus, termed the "activating" stimulus, triggers the oligomerization of the inflammasome [23–25]. The NLRP3 inflammasome can be triggered by multiple factors, comprising both endogenous and exogenous danger signals; e.g., K+ efflux, mitochondrial ROS, lysosome disruption [26–28], uric acid crystals, extracellular ATP, and bacterial toxins can begin this process [29,30]. These signals activate the NLRP3 inflammasome by inducing potassium efflux, mitochondrial dysfunction, and lysosomal damage, leading to the formation of the inflammasome complex and the activation of caspase-1 [28,31,32]. Extracellular ATP [33] and crystalline/particulate substances can result in lysosomal desta-

bilization [30,34]. The tripartite protein NLRP3 consists of an amino-terminal PYD, a central nucleotide-binding and oligomerization domain (NOD, also known as the NACHT domain), and a C-terminal LRR domain [35]. ASC is an adaptor protein that connects NLRP3 to the downstream effector protein. It contains a PYD domain at its N-terminus, enabling it to bind with the PYD domain of NLRP3, and a CARD domain at its C-terminus [36]. Pro-caspase-1 is an inactive form of caspase-1, an enzyme that contributes to the processing and secretion of proinflammatory cytokines. Following activation of the NLRP3 inflammasome, pro-caspase-1 is recruited to the complex via interactions with the CARD domain of ASC. Pro-caspase-1 is then cleaved and activated, producing mature IL-1β and IL-18 [37] (Figure 2).

Mammalian cell types and organelle architecture are essential for activating inflammasomes [38]. Different serotypes of Salmonella can trigger complex inflammatory responses as multiple host recognition systems need to be started [39]. From a biochemical perspective, exogenous ketone intake can play a role in the augmentation in NLRP3 inflammasome in monocytes [40]. Expression of inflammasome-related genes and IL-1β changes depending on the function of lithocholic acid (LCA) and other secondary bile acids that can be controlled by the microbiome inside the human body [41]. Deubiquitinases such as BRCC3 and JOSD2 facilitate deubiquitination and activation of the NLRP3-R779C variant. Individuals carrying this particular NLRP3 inflammasome variant may develop very early-onset inflammatory bowel disease [42].

### 3.3. The NLRP6 Inflammasome

The NOD-like receptor family pyrin domain-containing 6 (NLRP6), also known as PAN3, NALP6, PYPAF5, and CLR11.4 [43], exerts inflammasome-dependent as well as inflammasome-independent activity by activating caspases (caspase-1 or caspase-11) or by triggering NF-κB [44,45], the critical transcription factor of inflammatory processes. The physiological roles of NLRP6 comprise defense against pathogens, primarily regulating interactions between the microbiota and the intestinal mucosa, and proinflammatory and anti-inflammatory roles in tumorigenesis and neuroinflammation [45]. Depending on the specific tissue where it is activated, NLRP6 exerts distinct functions, making it an attractive therapeutic target [45]. In particular, NLRP6 is highly expressed in intestinal tissues, where it controls intestinal homeostasis and protects against inflammation-related colon tumorigenesis. The regulation patterns of NLRP6 are associated with inflammatory intestinal diseases such as Crohn's disease and ulcerative colitis [45,46]. In particular, NLRP6 acts as a sensor of bacterial and viral components. Following bacterial invasion, lipoteichoic acid (LTA), lipopolysaccharide (LPS), taurine, histamine, and spermine are the ligands identified in the direct and indirect modulation in NLRP6. LTA from Gram-positive bacteria and LPS from Gram-negative bacteria directly bind the LRR domain: the former induces the cleavage of caspase-11; contrarily, the latter triggers inflammasome complex formation with ASC and caspase-1 and the cleavage of pro-IL-1β and IL-18 [46]. Inflammasomes are positively modulated by taurine, which is microbiota-dependent, and the NLRP6-mediated secretion of IL-18 is suppressed by histamine and spermine [46]. NLRP6 participates in the anti-RNA virus immune response by recognizing long double-stranded RNA (dsRNA) and triggering interferon and interferon-stimulated genes (ISGs) via mitochondrial antiviral signaling proteins (MAVSs) [46].

### 3.4. The NLRP7 Inflammasome

The NLR family pyrin domain containing 7 (NLRP7) inflammasome is a close relative of NLRP2, but, in addition to inflammation processes, it is involved in embryonic development. NLRP7 mediates the recognition of bacterial lipopeptides in various immune cells, such as macrophages; promotes ASC-dependent caspase-1 activation, IL-1β, and IL-18 maturation; and hinders intracellular bacterial replication [47–49]. NLRP7 exerts a protective role in embryonic development and is expressed in nonimmune tissues such as the ovaries and oocytes [50]. As an example, the hydatidiform mole, a trophoblastic

disease resulting in a nonviable pregnancy, is the result of mutations in the NLRP7 encoding gene [47,50,51].

However, the mechanisms by which NLRP7 regulates embryonic development are still unclear [51]. Notable is the discovery that NLRP7 can even prevent inflammasome formation since it probably serves as a negative regulator of inflammation in quiescent cells but promotes inflammasome assembly and caspase-1 activation in response to a proper stimulus, such as infection [47].

### 3.5. The NLRP12 Inflammasome

The expression of the human NLR family pyrin domain containing 12 (NLRP12) was first identified in monocytes, macrophages, and granulocytes. During infections, NLRP12 serves as a cytosolic sensor that initiates the assembly of inflammasomes [52]. Initial investigations have linked truncated NLRP12 mutations to hereditary periodic fevers characterized by recurrent fevers, joint pain, and skin urticaria. Similarly, an NLRP12 single-nucleotide polymorphism has been associated with an increased risk of autoimmune and inflammatory diseases. Mutations in NLRP12 have also been linked to an increased risk of atopic dermatitis [53].

However, NLRP12 has both proinflammatory and anti-inflammatory functions independent of inflammasome formation. NLRP12 negatively regulates inflammation through the inhibition of inflammasome activation. Moreover, by forming complexes, NLRP12 inhibits the activation of the noncanonical NF-κB pathway and induces proteasome-mediated degradation of NF-κB-inducing kinase (NIK) [53]. During the inflammatory process, NLRP12 regulates immune cell migration by maintaining peripheral dendritic cells and neutrophils in a migration-competent state [53], and mice lacking NLRP12 are more likely to have colon inflammation, colorectal tumors, and atypical neuroinflammation [52].

Recent studies have revealed that NLRP12 is an inhibitor of autoinflammatory diseases and plays a role in a range of processes such as inflammatory bone mineral loss, arthritis, alcoholic liver injury-mediated apoptosis of hepatocytes, pathogen immune response, and the regulation of intestinal flora, such as in the maintenance of beneficial microbes involved in the inhibition of inflammation [54]. The most recent evidence highlights that heme release by red blood cell lysis during inflammatory disease, infection, or other cellular damages can activate a specific NLRP12-mediated PANoptosis, an innate immune inflammatory cell death pathway mediated by cell death-inducing complexes called PANoptosomes [52,55].

### 3.6. The NLRC4 Inflammasome

NLRC4 is a NOD-like receptor (NLR) family member. It consists of three main domains: a CARD at the N-terminus, a nucleotide-binding and oligomerization domain (NOD), and a leucine-rich repeat (LRR) domain (Figure 1). NLRC4 inflammasome is assembled in response to the detection of bacterial flagellin and components of the bacterial type III (T3SS) and type IV (T4SS) secretory system found on intracellular bacteria, including *Salmonella typhimurium*, *Shigella flexneri*, *Pseudomonas aeruginosa*, and *Legionella pneumophila* [56]. NLRC4 inflammasome does not require ASC for recruiting procaspase-1 because it comprises the CARD domain. Even though all NLRC4 inflammasomes do not require ASC, they do require interactions with the NLR family of apoptosis inhibitory proteins (NAIPs). NAIPs are receptors for bacterial protein ligands, and these interactions govern inflammasome specificity [57]. They function as direct receptors for bacterial flagellin, T3SS needle, and rod subunits. Ligand binding triggers the recruitment and activation of NLRC4 inflammasome [58]. Four NAIPs are present in mice, while only one NAIP is in humans. In mice, NAIP1 attaches to needle protein, NAIP2 detects rod proteins, and NAIP5/6 interacts with flagellin. NLRC4 might be specific to the needle protein rather than flagellin in human infectious diseases [57]. After ligand binding, NAIP receptors recruit bipartite adapter protein, which consists of a PYD and a CARD, through homotypic CARD–CARD interactions. CARD containing NLRC4 or ASC can recruit procaspase-1, followed by proximity-induced autoproteolytic activation within the complex. Such interactions

lead to the processing of pro-IL-1β and pro-IL-18 into their mature forms and GSDMD cleavage [58].

### 3.7. The AIM2 Inflammasome

The AIM2 inflammasome consists of an N-terminal PYD and a C-terminal HIN domain with tightly packed oligonucleotide or oligosaccharide binding folds. AIM2 inflammasomes can be formed when double-stranded microbial DNA is attached to relevant proteins [11], and AIM2-like receptors (ALRs) activate the inflammasome in response to specific pathogens. In mice, AIM2 inflammasomes play a pivotal role in initiating host defense against DNA viruses such as mouse cytomegalovirus and vaccinia virus [58]. AIM2 inflammasome triggers caspase-1 activation and subsequent cleavage of IL-1β and IL-18 against intracellular bacterial and viral infection. The AIM2 inflammasome is activated by the lysis of bacteria such as *Francisella tularensis* subsp. *novicida* or *Listeria monocytogenes* in the host cytosol [59,60]. The disruption of the nuclear membrane due to viral infection results in the release of double-stranded DNA (dsDNA). The AIM2 receptor binds to dsDNA through its HIN domain in the cytosol. AIM2 inflammasome can sense pathogen-derived cytosolic dsDNA and damaged or mislocalized DNA components. The AIM2 inflammasome does not comprise a CARD domain; therefore, it binds to ASC via PYD-PYD interaction. This inflammasome is formed through AIM2-ASC oligomerization. Pro-caspase-1 is bound through ASC by CARD–CARD interaction. In addition, the minimum DNA length to activate the AIM2 inflammasome is 80 base pairs long [57]. Enormous expression of AIM2 inflammasome may cause physical exacerbation such as psoriasis, abdominal aortic aneurysm, and systemic lupus erythematosus [61–63]. For instance, psoriasis occurs due to the autoinflammation by AIM2-mediated recognition of self-DNA in the cytoplasm of keratinocytes [62]. In contrast, reduced levels of AIM2 play a role in the progression of prostate and colorectal cancer [64,65].

### 3.8. The IFI16 Inflammasome

Similar to AIM2, the interferon-inducible protein 16 (IFI16) is an intracellular innate immune receptor that functions as a DNA sensor, but in contrast to AIM2, its role is expressed in the nucleus and proceeds toward the assembly of the inflammasome [66,67]. IFI16 inflammasome is specifically devised for the recognition of viruses replicating within the nucleus, thus providing an additional layer of complexity to innate intracellular immunity elucidation [47]. During abortive HIV infection, AIM2-induced IFI16 activates the inflammasome in CD4 T cells [68]. Inflammasomes are also activated in endothelial cells when they are infected by Kaposi sarcoma-associated herpesvirus (KSHV) [69]. However, the precise mechanisms of how IFI16 activates caspase-1 inflammasome still need to be fully understood, as the IFI16 pyrin domain does not bind with the ASC adapter. In addition, it has been observed that the oligomerization of IFI16 is insufficient for the induction of pyroptosis in THP1 cells [70].

### 3.9. The Pyrin Inflammasome

The pyrin inflammasome, known as the PYD-containing protein 3 (PYCARD) or the familial Mediterranean fever (FMF) inflammasome, is activated in response to specific triggers like bacterial toxins or stress signals. The best-characterized activator of the pyrin inflammasome is the Rho GTPase inhibitor toxin produced by certain bacteria, including the causative agent of FMF, called pyrin-associated autoinflammation with neutrophilic dermatosis (PAAND). Other stimuli, such as viral proteins and various stressors affecting the cellular cytoskeleton, can activate the pyrin inflammasome. Pyrin does not directly detect host-derived or pathogen-derived danger molecules. However, pyrin reacts to perturbations in cytoplasmic homeostasis caused by infection [71]. Pyrin comprises a PYD, two B-boxes, and a coiled-coil domain (Figure 1). Human pyrin includes a C-terminal B30.2 domain known as the PRY domain [58]. Following activation, pyrin binds with the adaptor protein ASC through homotypic Pyrin–Pyrin interactions. ASC contains a CARD

that allows the recruitment and activation of downstream effector protein caspase-1, which leads to the processing and release of IL-1β [72].

### 3.10. Non-Canonical Inflammasomes

While the canonical inflammasome pathway typically encompasses the activation of caspase-1, the non-canonical inflammasome pathway functions via a different mechanism. In the non-canonical inflammasome pathway, the activation of caspase-11 (in mice) or caspase-4 and caspase-5 (in humans) is central. Caspase-4/5/11 recognizes intracellular lipopolysaccharide (LPS) that is released from Gram-negative bacterial cell walls, and such occurrence induces GSDMD cleavage and pyroptosis [57]. Potassium efflux occurs via membrane pores due to GSDMD cleavage, which also triggers the activation of the NLRP3 inflammasome [56]. Then, the NLRP3 inflammasome initiates the processing of proinflammatory cytokines and releases IL-1β and IL-18 in response to non-canonical inflammasome activation [58].

## 4. Role of Inflammasomes in Inflammatory Diseases

Inflammasomes play a significant role in various inflammatory diseases [23] (Figure 3). The involvement of inflammasomes in a wide range of inflammatory diseases has been reported [23,73,74]. In this section, we discuss the role of inflammasome complexes in various inflammatory diseases, including multiple sclerosis (MS), experimental autoimmune encephalomyelitis, Alzheimer's disease (AD), Parkinson's disease (PD), atherosclerosis, type 2 diabetes (T2D), and obesity (Figure 3). Limited but emerging data suggest that inflammasomes may play a role in the development and progression of other inflammatory conditions, including rheumatoid arthritis (RA), inflammatory bowel disease (IBD), gout, psoriasis, and systemic lupus erythematosus (SLE) [75].

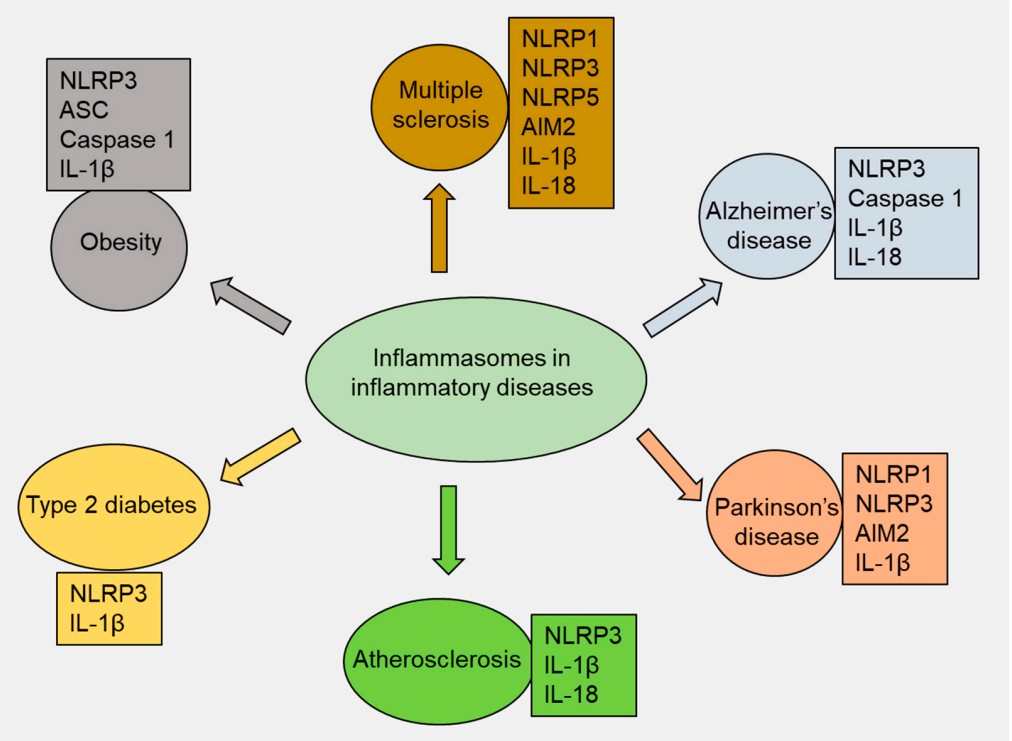

**Figure 3.** Inflammasomes in major inflammatory diseases. The figure presents an overview of the major inflammasomes activated in the indicated diseases and an overview of the major products of inflammasome activity.

### 4.1. Multiple Sclerosis

MS is a chronic inflammatory disease that influences the central nervous system (CNS) and is characterized by neurodegenerative symptoms and disability, for example, weakness, tiredness, incontinence, and paralysis. The immune system damages the myelin sheath that covers nerve fibers, leading to demyelination, inflammation, and axonal damage. Genetic and environmental factors are believed to be involved in the development of MS, although the exact cause of the disease remains unknown. There is no cure for MS, and the central focus of existing treatments revolves around using immunomodulatory medications to lessen the severity and frequency of relapses [76].

The role of inflammasomes in the pathogenesis of MS has been emphasized in recent years, with particular attention paid to the NLRP3 inflammasome. In MS, the NLRP3 inflammasome is activated in various cell types, including microglia, astrocytes, and CD4+ T cells [77]. Activation of the NLRP3 inflammasome leads to the release of IL-1β and IL-18, which promote inflammation and contributes to myelin destruction [78]. The NLRP3 inflammasome also regulates the blood–brain barrier (BBB), which is disrupted in MS, allowing immune cells to enter the CNS [79]. The role of NLRP3 in MS was supported by the observation that numerous small-molecule NLRP3 inhibitors can reduce disease severity. For example, MCC950 is a small-molecule NLRP3 inhibitor that binds to the NLRP3 NACDHT domain and blocks NLRP3 conformational changes and oligomerization [80]. Ketotifen, an antihistamine, has been shown to inhibit NLRP3 inflammasome action and reduce oxidative stress and the infiltration of T cells in the CNS [81]. IC100, a humanized antibody against the ASC component of inflammasomes, has also been developed to block NLRP3 [82]. These findings suggest that targeting the NLRP3 inflammasome is a promising approach for MS treatment.

Another inflammasome linked to MS is the AIM2 inflammasome. AIM2 is activated by cytosolic double-stranded DNA and leads to the maturation and release of IL-1β and IL-18, as well as caspase-1, which promotes the proteolytic cleavage of the cytokines, as mentioned earlier [83]. AIM2 is linked to the development of experimental autoimmune encephalomyelitis, an animal model of MS [84]. AIM2 inflammasome plays a role in the demyelination process in MS as it induces the expression of matrix metalloproteinases (MMPs) in microglia, a key contributor to the breakdown of the BBB [85].

Many inflammasomes linked to MS remain understudied. This list includes inflammasomes such as NLRP1, NLRC5, and others. NLRP1 is known to play a role in the activation of caspase-1 and the production of IL-1β and IL-18 in response to bacterial toxins, and multiple studies have also identified its involvement in numerous sclerosis cases [86,87]. Similarly, NLRC5 deficiency reduces the severity of experimental autoimmune encephalomyelitis, with lower inflammation and demyelination in the CNS [88]. As a result of these limitations, more research is needed to fully elucidate the role of inflammasomes in the pathogenesis of MS.

Experimental autoimmune encephalomyelitis (EAE) is a group of disorders characterized by axonal damage and demyelination of the CNS and has been used as a model for MS [88]. The EAE model leads to neurological symptoms, with motor deficits (from weakness and spasticity to complete paralysis) being the most prevalent. Additionally, sensory disturbances such as numbness, tingling, pain, and autonomic dysfunction affecting blood pressure, heart rate, and bladder and bowel function are commonly observed [89]. In EAE, the immune system is activated against self-antigens within the CNS, leading to the recruitment and activation of immune cells such as T cells and macrophages [90,91]. In addition to T cells and macrophages, B cells and natural killer (NK) cells have also been found to be implicated in the pathogenesis of EAE [92,93].

In EAE, inflammasomes are thought to be activated by DAMPs released by damaged or dying CNS cells. These DAMPs include extracellular ATP, uric acid crystals, and the high mobility group box (HMGB1) protein [94]. The existing literature shows that inflammasomes contribute to the pathogenesis of EAE primarily by promoting the activation of microglia [95]. Microglia, the resident immune cells of the CNS, play a vital role in

maintaining tissue homeostasis. However, they can become activated in response to injury or inflammation. Activated microglia generate proinflammatory cytokines such as IL-1β and IL-18, which further activate T cells and recruit other immune cells to sites of inflammation [96]. In addition to promoting microglial activation, inflammasomes can directly contribute to the production of proinflammatory cytokines. For example, the NLRP3 inflammasome promotes the activation of the NF-κB pathway, which leads to the production of proinflammatory cytokines, such as IL-1β and TNF-α, by microglia in EAE. These cytokines further activate T cells and promote the recruitment of other immune cells to sites of inflammation in the CNS, exacerbating tissue damage and disease progression [97]. Inflammasomes also contribute to the pathogenesis of EAE by activating other cell types in the CNS, such as astrocytes and oligodendrocytes [98,99]. AIM2 inflammasome activates astrocytes in EAE by producingIL-1β and other cytokines [98]. In addition, inflammasome activation via TNFR2 signaling can induce the expression of proinflammatory genes in oligodendrocytes, further exacerbating inflammation and contributing to tissue damage in EAE [99].

Inflammasomes also play a role in the demyelination process in EAE [100]. The *NLRP3* gene was found to be significantly upregulated in a demyelination model, and mice lacking this gene showed delayed neuroinflammation and demyelination and experienced loss of oligodendrocytes [100]. This effect was partly mediated by caspase-1 and IL-18 but not IL-1β. Interestingly, the lack of *NLRP3* did not result in delayed remyelination, unlike the absence of IL-1β. Inhibition of IL-18 may decrease demyelination but promote remyelination, suggesting that IL-18 is a potential therapeutic target for demyelinating diseases [100].

### 4.2. Alzheimer's Disease

AD is a complex and multifaceted disorder that affects millions of people worldwide. It is a progressive neurodegenerative disease that leads to cognitive impairment, memory loss, and behavioral changes. The pathological manifestations of AD are the accumulation of amyloid-beta (Aβ) plaques and tau tangles in the brain. These protein aggregates cause dysfunction and loss of neurons, leading to the death of brain cells [101]. Inflammation is a critical component of AD pathogenesis, and inflammasomes have emerged as crucial mediators of neuroinflammation in AD [102].

The NLRP3 inflammasome is the most extensively studied in AD. The NLRP3 inflammasome can be activated in response to various danger signals, such as Aβ and reactive oxygen species (ROS), leading to the activation of caspase-1 and the release of IL-1β and IL-18. Aβ can directly activate the NLRP3 inflammasome by binding to the purinergic receptor P2X7 on microglia, thus producing ROS and activating NLRP3 [103]. In addition, Aβ can induce lysosomal damage and the release of cathepsin B, a lysosomal protease, which can also activate the NLRP3 inflammasome [104].

Inflammasome activation in AD has been implicated in several pathological conditions with the production of proinflammatory cytokines, microglial activation, and the formation of Aβ plaques [105]. The function of microglia is the clearance of Aβ plaques, which accumulate in the brains of AD patients. In AD, microglia become chronically activated and cannot clear Aβ effectively [106]. In addition to the clearance of Aβ, microglia play an important role in regulating synaptic plasticity, a process essential for learning and memory [107]. Proinflammatory cytokines IL-1β and IL-18 induce the expression of amyloid precursor protein (APP) and beta-secretase 1 (BACE1), leading to increased Aβ production [108]. Moreover, inflammasome activation has been reported to induce the formation of Aβ oligomers and fibrils, which leads to the deposition of Aβ plaques [109].

In addition to their role in inducing neuroinflammation and Aβ deposition, inflammasomes have been implicated in other pathological processes, such as tau hyperphosphorylation and autophagy [110]. Tau hyperphosphorylation is a crucial pathological process in AD closely associated with the formation of neurofibrillary tangles (NFTs), a hallmark of AD pathology. Tau protein is usually involved in stabilizing microtubules in neurons.

However, when it becomes hyperphosphorylated, it detaches from microtubules and aggregates into NFTs, disrupting axonal transport and neuronal dysfunction [111]. Recent pieces of evidence have suggested that the activation of inflammasomes (such as NLRP3) can induce tau hyperphosphorylation by promoting the expression levels of tau kinases, such as glycogen synthase kinase-3β (GSK3β) and cyclin-dependent kinase 5 (CDK5), and inhibiting the activity of tau phosphatases, such as protein phosphatase 2A (PPP2A) and protein phosphatase 1 (PP1) [112,113]. ROS produced by activated microglia and neurons can also induce tau hyperphosphorylation by activating the NLRP3 inflammasome and promoting the expression levels of tau kinases [114].

Inflammasome activation can also suppress autophagy, a cellular mechanism that clears misfolded proteins and damaged organelles, leading to the buildup of toxic protein aggregates and cellular debris [115]. Autophagy impairment has been shown to induce tau hyperphosphorylation by increasing tau kinase levels and decreasing tau phosphatase activity [116]. NLRP3 inflammasome activation can inhibit autophagy through several mechanisms, including lysosomal damage, impairing lysosomal acidification, and reducing lysosomal enzyme activity [115].

Inflammasome activation has been reported to exacerbate cognitive impairment and Aβ pathology in transgenic AD mice [117]. Inhibition of inflammasome activation using pharmacological or genetic approaches improved cognitive function and reduced Aβ pathology in AD mice. For example, treatment with MCC950, a selective NLRP3 inhibitor, has reduced Aβ pathology and enhanced cognitive function in AD mice [118]. Similarly, deletion of NLRP3 or caspase-1 was associated with reduced Aβ pathology and improved cognitive function in AD mice [119].

Inflammasome activation in peripheral tissues, including the gut and liver, has also been implicated in AD pathogenesis by inducing systemic inflammation and impaired brain function [120]. For example, gut dysbiosis, characterized by an imbalance in the gut microbiota, has been demonstrated to induce inflammasome activation and increase Aβ deposition in the brain [121]. Similarly, liver dysfunction, which commonly occurs in AD patients, can induce inflammasome activation and increase the production of proinflammatory cytokines and oxidative stress, causing systemic inflammation and cognitive impairment [122].

Other inflammasomes, such as NLRP1, AIM2, and NLRC4, have also been suggested to contribute to neuroinflammation in AD, although their roles are less well-established than those of NLRP3.

In experimental models of AD, NLPR1 seems able to promote neuronal pyroptosis [123], whereas AIM2 deletion mitigates Aβ deposition and microglial activation but increases the expression of inflammatory cytokines [124].

In another in vivo experimental model, a significant increase in the expression level of the NLRC4 inflammasome, ASC, IL-1β, and p-Tau protein-positive cells after pharmacologic treatment for the induction of an Alzheimer's-like disease in animals has been found [125]. In contrast, no significant difference was seen in other inflammasome components such as NLRP1, NLRP3, AIM2, IL-18, and caspase-1. These findings suggest that the NLRC4 inflammasome is involved in the typical neuroinflammation and memory impairment of AD [125].

### 4.3. Parkinson's Disease

PD is a chronic neurodegenerative disorder characterized by progressive loss of pigmented nigrostriatal dopaminergic neurons in the substantia nigra pars compacta (SNpc) region of the brain. The disease's progression leads to motor symptoms such as tremors, rigidity, and bradykinesia [126]. Activated glial cells, which compose most of this inflammatory response, contribute to this neurodegenerative process by producing toxic molecules [127]. Various factors, including glial reactions, T-cell infiltration, and increased expression of inflammatory cytokines, trigger inflammatory responses in PD. However, the NLRP3 inflammasome is the most widely studied in the pathogenesis of PD [128].

NLRP3 inflammasome activation is a two-step process, i.e., priming and activation [25]. Peripheral inflammation can transform primed microglia into a state that can trigger more robust neurodegenerative responses [129]. Although the precise mechanism of inflammasome priming and activation in PD has not been fully elucidated, emerging research data suggest the involvement of cytokines such as IL-1β and TNF-α [130]. When these cytokines are secreted by activated glia in the brain or are present in circulating blood, the permeability of the BBB increases, and the expression levels of cellular adhesion molecules, such as selectins, are upregulated in microvascular endothelial cells [131].

Various other factors have been suggested to activate the NLRP3 inflammasome in PD, such as oxidative stress, mitochondrial dysfunction, and the accumulation of misfolded proteins such as alpha-synuclein (α-syn) [132]. α-syn is a presynaptic protein typically involved in the regulation of neurotransmitter release. In PD, α-syn accumulates in insoluble aggregates that are the principal constituents of Lewy bodies—pathological hallmarks of PD. The misfolded proteins can activate NLRP3 inflammasome by triggering lysosomal damage, leading to the release of cathepsin B, which then initiates the inflammasome assembly [133]. In addition, α-syn can activate NLRP3 by interacting with TLR2, leading to the activation of downstream signaling pathways [134].

In addition to the NLRP3 inflammasome, other inflammasome complexes, such as the AIM2 and the NLRP1 inflammasomes, have also been implicated in the pathogenesis of PD [7,135,136]. Activation of these inflammasomes induces the production of IL-1β and IL-18, which can contribute to the neuroinflammatory response in PD [135]. The AIM2 inflammasome is activated by double-stranded DNA in the cytoplasm, and research findings have indicated that it is activated in response to α-syn accumulation in PD [7]. Specifically, research has shown that extracellular α-syn can be taken up by microglia and transported to the cytoplasm, where it can activate the AIM2 inflammasome [137]. In PD, the activation of the NLRP1 inflammasome has been reported to be associated with the accumulation of α-syn and the induction of neuronal cell death [136]. Considering the involvement of inflammasomes in PD pathology, targeting the inflammasome complex or downstream inflammatory pathways would be an ideal therapeutic approach to mitigate neuroinflammation and slow down PD progression.

### 4.4. Atherosclerosis

Atherosclerosis is a chronic inflammatory disease characterized by the deposition of lipids, immune cells, and extracellular matrix (ECM) within the arterial walls [138]. Inflammation is vital in initiating, progressing, and rupturing atherosclerotic plaques, which can lead to cardiovascular events, including myocardial infarction and stroke [139]. Both innate and adaptive immunity play a critical role in the initiation, progression, and destabilization of atherosclerotic plaques and dyslipidemia [140]. Innate immune cells, including monocytes, macrophages, and dendritic cells, may accumulate in the arterial wall, contributing to the inflammatory response [141]. CD31+ endothelial cells and CD68+ macrophages within atherosclerotic lesions in human carotid arteries exhibit significant levels of the purinergic 2X7 receptor (P2X7R) [142]. P2X7R has been reported to be involved in the progression of atherosclerosis by inducing the activation of the NLRP3 inflammasome [143]. Macrophages are among the first immune cells to accumulate in the plaque, which contributes to the uptake of ox-LDL and the formation of foam cells [144]. Foam cells are lipid-laden macrophages that play a significant role in the progression of plaque formation and the narrowing of the arteries [142].

It has been reported that NLRP3 inflammasome activation in macrophages within atherosclerotic plaques can promote the formation of foam cells [145]. Sterol regulatory element-binding protein 1 (SREBP-1) is a transcription factor critical in regulating lipid metabolism. Varghese et al. demonstrated that the activation of the NLRP3 inflammasome was facilitated by SREBP-1, leading to the formation of macrophage foam cells induced by ox-LDL [146]. The buildup of ox-LDL is crucial in the development of atherosclerotic plaques [147].

Inflammasomes are activated in response to various stimuli, including cholesterol crystals and ox-LDL, producing and secreting proinflammatory cytokines such as IL-1β and IL-18 [146,148]. These cytokines play a dominant role in the development of atherosclerosis by triggering the recruitment and activation of immune cells, inducing the expression of adhesion molecules on endothelial cells and stimulating smooth muscle cell proliferation and ECM deposition [149]. IL-1β is one of the most potent mediators of atherosclerosis and is involved in various phases of the disease. IL-1β can induce the expression of adhesion molecules on endothelial cells, resulting in the recruitment of monocytes into the arterial wall. It can also activate smooth muscle cells and promote their migration, proliferation, and matrix deposition, leading to fibrous cap formation [150]. IL-18 is another proinflammatory cytokine produced by the inflammasome and has been linked to the pathogenesis of atherosclerosis [151]. IL-18 activates T lymphocytes, which play an important role in the adaptive immune response in atherosclerosis [152]. IL-18 can also promote the development of atherosclerosis by activating endothelial cells and increasing the expression of adhesion molecules, which helps the recruitment of immune cells into the arterial walls. Furthermore, IL-18 can induce the production of other proinflammatory cytokines, including IL-6 and TNF-α, which contribute to plaque development and destabilization [153].

The TLR4, NF-κB, and JAK/STAT pathways are all involved in inflammatory signaling and play an essential role in atherosclerosis. TLR4, which is expressed in immune cells, can detect a range of ligands, such as LPS, oxLDL, and HMGB1 [154]. Following activation, TLR4 can initiate a signaling cascade that, ultimately, activates NF-κB and produces proinflammatory cytokines [155]. The JAK/STAT pathway can activate proinflammatory pathways in response to cytokines such as IL-6 [156]. Furthermore, considering the role of inflammation in atherosclerosis, traditional risk factors such as high LDL levels, obesity, angiotensin II, and smoking remain influential in the development of atherosclerosis [157].

### 4.5. Type 2 Diabetes

Type 2 diabetes (T2D) is a metabolic condition characterized by insulin resistance, which leads to hyperglycemia and glucose intolerance [158]. Nearly 90% of diabetic patients exhibit insulin resistance [159]. In recent years, there has been a growing interest in the role of inflammation in the development of this disease. Studies suggest that subclinical chronic inflammation and innate immune system activation are vital pathogenetic elements in the emergence of insulin resistance and T2D [160].

The NLRP3 inflammasome and the IL-1β pathway have been reported in T2D [161,162]. According to a study by Lee et al., NLRP3 inflammasome activation was elevated in myeloid cells obtained from individuals with T2D [162]. Studies have demonstrated that oligomers of islet amyloid polypeptide (IAPP), a protein associated with the formation of amyloid deposits in the pancreas during T2D, can activate the NLRP3 inflammasome, leading to the production of mature IL-1β [163]. The involvement of NLRP3 in T2D was further documented by the observation that silencing the *NLRP3* gene showed a significant improvement in the development of diabetic cardiomyopathy (DCM) in a rat model of T2D [164]. Furthermore, rosuvastatin demonstrated alleviation of diabetic cardiomyopathy in a rat model of T2D by inhibiting the NLRP3 inflammasome [165]. γ-Tocotrienol (γT3) also effectively slowed down the advancement of T2D by inhibiting the NLRP3 inflammasome [165]. These compelling data strongly support the pivotal role of inflammasomes in the pathogenesis of T2D.

Inflammasomes have also been implicated in the development of β-cell dysfunction [166,167]. β cells are the cells in the pancreas responsible for insulin production and secretion, and their dysfunction is a prominent feature of T2D [168]. Chronic activation of inflammasomes produces ROS and oxidative stress, which can damage beta cells and impair their function. This process can lead to a decrease in insulin secretion and the development of hyperglycemia [169]. Indeed, Sokolova et al. reported that the deletion of the NLRP3 inflammasome was associated with increased β-cell function and viability in the presence of hypoxia and oxidative stress [167].

Chronic inflammation is a signature characteristic of T2D, and the primary molecular links between inflammation and T2DM are macrophage mediators TNF-α, IL-1β, and IL-6. These inflammatory cytokines are generated by activated macrophages and adipocytes in adipose tissue and are elevated in the serum of individuals with insulin resistance and T2D [170]. Inflammatory mediators such as TNF-α, IL-1β, and IL-6 can stimulate insulin resistance by interrupting insulin signaling in peripheral tissues through activation of various inflammatory pathways, including NF-κB and c-JUN N-terminal kinase (JNK) pathways [171]. These pathways interfere with the insulin signaling cascade by inducing the phosphorylation of serine residues on IRS proteins, which results in their degradation and, ultimately, leads to insulin resistance [172]. Several drugs with anti-inflammatory properties have been reported to lower both acute-phase reactants and glycemia and decrease the risk of developing T2D. These drugs include thiazolidinediones (TZDs) and glucagon-like peptide-1 (GLP-1) receptor agonists. These drugs can target various components of the inflammatory signaling pathways involved in insulin resistance and have been shown to improve insulin sensitivity and glycemic control in individuals with T2D [173]. For example, pioglitazone is an oral antidiabetic from the thiazolidinedione drug class and is best known for its dual agonist activity on both PPAR-γ and PPAR-α. Pioglitazone can mitigate diabetic renal damage by inhibiting the activation of the renal AGE/RAGE axis and reducing NF-κB expression [174]. This effect was associated with decreased NLRP3 levels and subsequent reduction in the secretion of inflammatory cytokines [174]. Liraglutide, a GLP-1 receptor agonist, has been reported to improve the disease score in EAE mice, associated with the downregulation of the NLRP3 pathway [175].

*4.6. Obesity*

Obesity is a central component of metabolic syndrome, characterized by excessive fatty tissue expansion induced by immune cell infiltration, particularly macrophages, and adipocyte hypertrophy [176]. The infiltration of immune cells into adipose tissues can be exacerbated by excessive consumption of fat and other macronutrients without sufficient antioxidant intake [177]. Recent studies using a bidirectional Mendelian randomization approach have provided compelling evidence that higher levels of adiposity caused by fat mass, obesity-associated genes, and single nucleotide polymorphisms (SNPs) of melanocortin receptor 4 are intricately connected with elevated levels of the inflammatory marker CRP [178]. In particular, the NLRP3 inflammasome plays an essential role in the pathogenesis of obesity by increasing adiposity, insulin resistance, glucose intolerance, and inflammation [179]. It has been reported that the knockout of the NLRP3 inflammasome can protect against obesity-induced pathologies, making it a viable target for therapeutic intervention [180]. NLRP3 inflammasome activation in adipose tissue is associated with the recruitment of macrophages, production of proinflammatory cytokines, and induction of insulin resistance [13]. Inflammasome-deficient mice are protected from developing obesity and insulin resistance when fed a high-fat diet [181]. In addition, inhibiting NLRP3 inflammasome activation in obese mice improved glucose homeostasis and insulin sensitivity [182]. Fatty acid accumulation in obese adipose tissue can increase ceramide production, a danger signal to stimulate the formation of the NLRP3 inflammasome complex [183].

Excessive intake of calories can result in the infiltration of macrophages into adipose tissue and the production of proinflammatory cytokines. However, deficiencies in NLRP3, ASC, and caspase-1 have been shown to offer protection against obesity-induced insulin resistance and metabolic dysfunction [184]. The ASC adaptor protein is an essential component of the inflammasome complex and acts as a bridge between NLRP3 and caspase-1 [185]. However, ASC-induced specks are not a prerequisite for inflammation activation but do maximize IL-1β processing [186]. Caspase-1 activation promotes the cleavage of pro-IL-1β and pro-IL-18 into their mature forms [187]. Another vital pathway implicates the activation of toll-like receptors (TLRs), which recognize microbial and endogenous ligands. TLR activation can produce proinflammatory cytokines and activate the inflammasome complex [154].

In addition to the direct activation of the inflammasome complex, several other pathways have been implicated in the regulation of inflammasome activation in obesity. These include the production of ROS, which can promote NLRP3 inflammasome activation, and the modulation of lipid metabolism, which can contribute to the accumulation of lipid intermediates that activate the inflammasome complex [188]. Furthermore, several adipokines and hormones, including leptin and insulin, have been shown to regulate inflammasome activation in obesity. Leptin and insulin are essential hormones in regulating energy metabolism and developing obesity. Adipocytes produce leptin and act on the hypothalamus to control food intake and energy expenditure [189]. Conversely, insulin is produced by the pancreas and regulates glucose uptake and metabolism in various tissues, including adipose tissue [190]. Leptin has been shown to promote inflammasome activation by inducing the generation of ROS and activating the NLRP3 inflammasome [191]. In addition, leptin enhances the secretion of proinflammatory cytokines, including IL-1β and IL-6, by macrophages in adipose tissue [192]. IL-1β and IL-6 can induce insulin resistance and promote the development of metabolic dysfunction in obesity. On the other hand, insulin has been shown to have both proinflammatory and anti-inflammatory effects on the inflammasome complex. Insulin can activate the NLRP3 inflammasome by promoting the generation of ROS and activating the TLR4 signaling pathway [193]. However, it has been observed that insulin can also inhibit inflammasome activation by inhibiting the secretion of proinflammatory cytokines and promoting the secretion of anti-inflammatory cytokines such as IL-10 [194].

### 4.7. Other Inflammatory Diseases

Emerging evidence suggests that inflammasomes have a potential role in the development and progression of other inflammatory conditions, which include RA, IBD, gout, psoriasis, and SLE [75]. As a prototypical autoimmune disease, RA is primarily characterized by inflicting damage to the bones and cartilage. In a study conducted by Guo et al., it was observed that the NLRP3 inflammasome exhibited significant activation in the synovial tissue of RA patients and mice with collagen-induced arthritis (CIA) [195]. Activation of the inflammasome in synovial cells promotes the secretion of IL-1β and IL-18, contributing to the chronic inflammation, joint damage, and cartilage destruction observed in RA [196]. IBD is a group of chronic inflammatory disorders primarily affecting the gastrointestinal tract. The two main types of IBD are ulcerative colitis (UC) and Crohn's disease (CD). UC is characterized by inflammation and ulcers typically limited to the colon's inner lining (large intestine) and rectum. Conversely, CD can affect any part of the gastrointestinal tract, from the mouth to the anus. The exact etiology of IBD remains uncertain; however, available evidence indicates a complex interaction between genetic and environmental factors. The activation of inflammasomes, particularly NLRP3, has been shown to facilitate the secretion of proinflammatory cytokines and the recruitment of immune cells, contributing to tissue damage and the progression of IBD [197]. Gout is a type of inflammatory arthritis that occurs when uric acid crystals accumulate in the joints. When the uric acid level becomes too high, sharp urate crystals can form in the joints, triggering an inflammatory response and causing the characteristic symptoms of gout. The role of inflammasomes, particularly NLRP3, has been implicated in the pathogenesis of gout [30,198]. Studies indicate that both monosodium urate (MSU) and calcium pyrophosphate dihydrate (CPPD) crystals can activate the caspase-1-activating NALP3 inflammasome, leading to the production of active IL-1β and IL-18 [30]. Psoriasis is a chronic autoimmune skin condition that leads to the rapid buildup of skin cells. It is described by red, thickened patches of skin covered with silvery scales. The exact cause of psoriasis is not entirely understood, but evidence indicates the involvement of a combination of genetic, immune system, and environmental factors. Experimental data suggest that various inflammasomes, including NLRP1, NLRP3, and AIM2, contribute to the development and progression of psoriasis [199,200]. A study by Verma et al. showed that the activation of NLRP3 inflammasomes in psoriasis patients through TNF-α was evidenced by the observation that anti-TNF therapy normal-

ized plasma IL-1β and IL-18 levels as well as caspase-1 reactivity [201]. Furthermore, in a study conducted by Tervaniemi et al., it was observed that the keratinocytes of psoriatic skin contain various components of the active inflammasome, namely NOD2, PYCARD, CARD6, and IFI16 [202]. Systemic lupus erythematosus, commonly known as lupus, is a chronic autoimmune disease that can affect multiple organs and systems in the body. The exact cause of SLE is unknown, but the involvement of a combination of genetic, hormonal, and environmental factors has been suggested. In SLE, the immune system becomes over-active and mistakenly attacks healthy tissues, causing inflammation and damage. Available experimental studies have shown a correlation between *NLRP1*, *NLRP3*, and *IL1B* genes and SLE, either as susceptibility factors or their influence on disease severity [203–206]. Furthermore, a study by Ma et al. suggested that the expression of the NEK7-NLRP3 complex may exhibit a protective role in developing SLE and is inversely associated with disease activity [207].

## 5. Inflammasomes as Therapeutic Targets in Inflammatory Diseases

Several compounds have been identified as potential inhibitors or modulators of the NLRP3 inflammasome, which is the most well-known and well-characterized inflammasome complex. These compounds can specifically inhibit the NLRP3 inflammasome assembly or its downstream pathways. This section discusses compounds that target NLRP3 and other inflammasomes in inflammatory diseases (Table 1).

**Table 1.** Therapeutic compounds targeting inflammasome complexes in inflammatory diseases.

| Therapeutic Agents | Diseases | Targeted Inflammasomes | Functions |
|---|---|---|---|
| MCC95 (CP-456773, CRID3) | MS | NLRP3, IL-1β | MCC950 could reduce clinical symptom of MS [208]. |
| | AD | NLRP3 | MCC950 reduced Aβ pathology and improved cognitive function in AD mice [118]. |
| | Gout | IL-1β | MCC950 significantly reduced the production of IL-1β and neutrophil infiltration in the inflamed joint [209]. |
| | Atherosclerosis | NLRP3, ASC, Caspase-1, GSDMD-N, IL-1β, IL-18 | MCC950 treatment reduced plaque areas and macrophage contents [210]. |
| | Diabetic encephalopathy | NLRP3, ASC, caspase-1, IL-1β | MCC950 treatment improved insulin sensitivity in db/db mice, thereby alleviating diabetic encephalopathy [211]. |
| | Diabetic nephropathy | NLRP3, caspase-1, IL-1β | MCC950 treatment reduced kidney injury in diabetic nephropathy [212]. |
| Ketotifen (Zaditor®) | MS | NLRP3 | Ketotifen treatment reduced both the prevalence and severity of EAE disease [81]. Ketotifen restored balance of oxidative stress, and reduced infiltration of T cells in the CNS [81]. |
| IC100 (IgG4) | MS | ASC | IC100 decreased the trafficking of CD4+, CD8+ T cells, and CD11b + MHCII+ cells into the CNS [82]. IC100 treatment reduced the number and activation state of CNS resident microglia [82]. |
| Fenamate NSAIDs (flufenamic acid and mefenamic acid) | AD | NLRP3 | Fenamate NSAIDs showed therapeutic benefits in a model of memory loss caused by amyloid beta and in a transgenic mouse model of AD [213]. |

**Table 1.** *Cont.*

| Therapeutic Agents | Diseases | Targeted Inflammasomes | Functions |
|---|---|---|---|
| Tranilast (N-[3′,4′-dimethoxycinnamoyl]-anthranilic acid) | Diabetic nephropathy | NLRP3 | Tranilast effectively decreased urinary albumin excretion, a significant clinical indicator of diabetic nephropathy [214]. |
| | Gestational diabetes mellitus | NLRP3, TNF-α, IL-6 | Tranilast exhibited a significant amelioration of GDM symptoms in mice [215]. |
| | T2D | NLRP3 | Tranilast showed therapeutic effects in mouse models of T2D [216]. |
| | Gouty arthritis | NLRP3 | Tranilast showed therapeutic effects in mouse models of gouty arthritis [216]. |
| | Atherosclerosis | NLRP3 | Tranilast showed notable efficacy in ameliorating vascular inflammation and reducing atherosclerosis in both low-density lipoprotein receptor-deficient and apolipoprotein E-deficient mouse models [217]. |
| Canakinumab (ILARIS®) | Gouty arthritis | NLRP3, IL-1β | Canakinumab is used in the treatment of gouty arthritis [218]. Canakinumab is effective in reducing the number of gout flares in patients with a history of gout [219]. |
| Glyburide | Gouty arthritis, silicosis, and AD | NLRP3 | Glyburide is thought to be effective against conditions like gouty arthritis, silicosis, and AD, where excessive IL-1β production via Cryopyrin-dependent pathways plays a significant role in the pathology [220–222]. |
| Pioglitazone | Diabetes mellitus | NLRP3 | Pioglitazone is effective against diabetic renal damage [174]. |
| Liraglutide | EAE | NLRP3 | Liraglutide treatment improves the disease score in EAE mice [175]. |
| Rosuvastatin | Diabetic cardiomyopathy | NLRP3 | Rosuvastatin induced reduction of diabetic cardiomyopathy in a rat model of T2D [165]. |
| γ-Tocotrienol | T2D | NLRP3 | γ-Tocotrienol (γT3) is effective in slowing down the advancement of T2D [165]. |

MCC950 (CP-456773, CRID3) is a potent small molecule inhibitor that mainly targets the NLRP3 inflammasome. It binds to the inflammasome's nucleotide-binding domain to stop its activation [223]. MCC950 has been tested in several preclinical models of inflammatory diseases, such as MS, gout, AD, atherosclerosis, and acute lung injury [209,224]. Using an EAE model as their disease model for MS, Xu et al. discovered that treatment with MCC950 in conjunction with rapamycin might lessen both the clinical symptoms and the release of cytokines (such as IL-1β) in immune cells [208]. These results support the importance of the NLRP3/IL-1β axis in the pathogenesis of MS. Furthermore, MCC950 treatment can reduce Aβ pathology and improve cognitive function in AD mouse models [118]. In a mouse model of gout, MCC950 drastically reduced the production of IL-1β and neutrophil infiltration in the inflamed joint [209]. A longitudinal follow-up study revealed that patients with spinal cord injury (SCI) are more likely to develop AD [225]. MCC950 has been reported to alleviate the inflammatory response and promote functional recovery in an acute mouse model of spinal cord injury (SCI) [226]. The observed effect of MCC950 was partly mediated by inhibition of NLRP3 inflammasome complexes assembly, such as NLRP3-ASC and NLRP3-Caspase-1, as well as by reducing the release of TNF-α, IL-1β, and IL-18 [226]. For instance, selective NLRP3 inhibitors such as MCC950 have been shown to reduce Aβ pathology and improve cognitive function in AD mice [118].

A significant reduction in plaque size observed in a hyperlipidemic mouse model after the administration of MCC950 [227] indicates that MCC950 is a promising therapeutic candidate for atherosclerosis. Recently, Zeng et al. evaluated the efficacy of MCC950 in treating

atherosclerosis in a mouse model. The study used an intraperitoneal injection of MCC950 to 8-week-old apoE−/− mice fed a high-fat diet for 12 weeks. Administration of MCC950 resulted in a reduction of the plaque areas and macrophage contents [210]. Mechanistic analysis showed that MCC950 effectively inhibited the activation of the NLRP3/ASC/Caspase-1/GSDMD-N axis and restrained the production of IL-1β and IL-18 in both aorta and cell lysates [210].

Zai et al. reported that treatment with MCC950 markedly improved insulin sensitivity and reduced diabetic encephalopathy in db/db mice [211]. In their study, it was observed that the hippocampus of diabetic db/db mice exhibited higher expression levels of inflammasome components, including NLRP3, ASC, and caspase-1, as well as IL-1β [211]. Treatment with MCC950 reversed the increased expression levels of NLRP3, ASC, and IL-1β and increased caspase-1 activity in the hippocampus [211]. Furthermore, the study revealed that the administration of MCC950 to db/db mice ameliorated anxiety- and depression-like behaviors and improved cognitive dysfunction [211]. These results propose that suppression of NLRP3 inflammasome activation may be a potential therapeutic approach for diabetic encephalopathy. Zhu et al. conducted a study in which they administered intraperitoneal injections of MCC950 (10 mg/kg) twice per week for 8 weeks to 12 weeks in type 2 diabetic db/db mice to examine the effectiveness of MCC950 in the setting of diabetic nephropathy [212]. Mice administered MCC950 showed a notable improvement in kidney injury in diabetic nephropathy, which was mediated partly by inhibition of the NLRP3/caspase-1/IL-1β axis [212].

Ketotifen (Zaditor®) is an antihistamine drug that is used to treat allergic rhinitis (hay fever) and allergic conjunctivitis. Ketotifen has been shown to suppress NLRP3 inflammasome activity in animal models of EAE [81]. In a study using C57BL/6 mice, EAE was induced by immunization with $MOG_{35-55}$ [81]. The mice received daily injections of ketotifen from the 7th to the 17th day after disease induction. Mice that received this early intervention with ketotifen showed a substantial reduction in both the prevalence and severity of the disease [81]. The protective effect was related to decreased NLRP3 inflammasome activation, improved oxidative stress balance, and reduced T-cell infiltration in the CNS [81].

A humanized antibody called IC100 (IgG4) has been specifically developed to target the ASC component of inflammasomes. Desu et al. investigated the functional and immunological effects of IC100 in an EAE animal model of MS [82]. In the EAE model of MS, the disease was induced by immunizing C57BL/6 mice with $MOG_{35-55}$. The mice were then treated with either a vehicle or increasing doses of IC100 (10, 30, and 45 mg/kg). In their study, Desu et al. observed that treatment with IC100 reduced the trafficking of CD4+, CD8+ T cells, and CD11b + MHCII+ cells into the CNS [82]. Moreover, IC100 management alleviated the number and activation state of CNS resident microglia [82]. Based on these findings, using a monoclonal antibody to target ASC can dramatically alter innate and adaptive immune responses in the MOG-induced EAE model, resulting in improved clinical outcomes.

It is worth noting that animal models often do not mirror human diseases. More translational studies and clinical trials are required to determine whether these therapeutic strategies can be applied to humans.

A study by Daniels et al. showed that several fenamate-class nonsteroidal anti-inflammatory drugs (NSAIDs) that are clinically approved displayed selectivity in inhibiting the NLRP3 inflammasome [213]. This inhibition occurs by targeting the volume-regulated anion channel (VRAC) of macrophages [213]. Flufenamic acid and mefenamic acid, which belong to the fenamate class of drugs, are potent in rodent models of inflammation in the air pouch and peritoneum [213]. Furthermore, fenamatos showed therapeutic benefits in an amyloid beta-induced amnesia model and a transgenic mouse model of AD [213].

Tranilast (N-[3′,4′-dimethoxycinnamoyl]-anthranilic acid), an anti-inflammatory agent mainly used in treating allergic disorders, has been assessed in preclinical and clinical

studies for diverse inflammatory conditions. Tranilast efficiently reduces urinary albumin excretion, a significant clinical indicator of diabetic nephropathy, by targeting the NLRP3 inflammasome pathway [214]. The protective effect of tranilast on gestational diabetes mellitus (GDM) was examined using a genetic GDM mouse model in a recent study conducted by Cao and Peng in 2022 [215]. Pregnant C57BL/KsJdb/+ (db/+) female mice served as GDM mice and were orally administered a daily dose of 20 mg/kg tranilast or metformin for two weeks [215]. Tranilast significantly ameliorated GDM symptoms in mice [215]. Additionally, tranilast showed a remarkable decrease in the higher expression of NLRP3, TNF-$\alpha$, and IL-6 [215]. These data propose that tranilast holds promise as a possible therapeutic intervention for GDM. A study by Huang et al. found that tranilast significantly prevented and treated human diseases associated with NLRP3 inflammasome activation in mice. These diseases include gouty arthritis, cryopyrin-associated autoinflammatory syndromes, and T2D [216]. Mechanistically, tranilast demonstrates inhibitory effects on NLRP3 inflammasome activation in macrophages but has no impact on AIM2 or NLRC4 inflammasome activation [216]. Tranilast also disrupted the development of the NLRP3 inflammasome by promoting NLRP3 ubiquitination, thereby decreasing the inflammatory response [217]. In another study, tranilast exhibited remarkable efficacy in ameliorating vascular inflammation and reducing atherosclerosis in both low-density lipoprotein receptor-deficient and apolipoprotein E-deficient mouse models [217].

Anthocyanins are natural pigments found in various fruits and vegetables, such as blueberries, raspberries, and red cabbage. These compounds have been exhibited to contain potent anti-inflammatory properties by restraining the activation of NLRP3 inflammasome [228]. Anthocyanins act by subduing the activation of the NLRP3 inflammasome through numerous mechanisms, including the inhibition of NF-$\kappa$B signaling, decreasing the expression of proinflammatory cytokines, and suppressing oxidative stress [229]. In the context of nonalcoholic fatty liver disease (NAFLD), upregulation of the NLRP3 inflammasome has been related to the disease's development [230]. This occurs when hepatic cells are exposed to danger signals, such as saturated fatty acids or cholesterol, which trigger the inflammasome and stimulate the release of proinflammatory cytokines. Moreover, anthocyanins have been shown to improve insulin sensitivity and lipid metabolism in NAFLD patients, further highlighting their therapeutic promise [231].

Other natural compounds, including oridonin, have been reported to hinder NLRP3 inflammasome in inflammatory conditions like dextran sulfate sodium-induced colitis [232,233]. Oridonin is derived from the *Rabdosia rubescens* plant. Quercetin, a natural compound abundantly found in various fruits and vegetables, has been reported to hinder the oligomerization of ASC and effectively prevent IL-1-mediated mouse vasculitis [234]. A previous study showed that quercetin could obstruct the initiation of the NLRP3 inflammasome in epithelial cells when triggered by Escherichia coli O157:H7 [235]. These data indicate that both oridonin and quercetin possess significant potential for future research exploring their effectiveness for treating various other inflammatory conditions.

Other therapies targeting inflammasome components have been assessed in the preclinical and clinical settings of diverse inflammatory conditions. For instance, anakinra (KINERET®), a recombinant form of the naturally occurring IL-1 receptor antagonist, binds to the same receptor as IL-1$\beta$ but does not activate it [236]. Anakinra competitively hinders the binding of IL-1$\beta$ to the receptor, thereby obstructing its proinflammatory effects. Two clinical studies have reported that IL-1 blockade exhibits promise as an efficient therapy for acute gouty arthritis [237,238]. The role of IL-1$\beta$, a key cytokine involved in driving inflammatory responses, has been investigated by Bertoni et al. The researchers studied circulating monocytes from patients with COVID-19 and observed the presence of ASC specks. These ASC specks were found to colocalize with the NLRP3 inflammasome and were associated with the spontaneous secretion of IL-1$\beta$ in vitro [239]. Notably, the study revealed that this spontaneous inflammasome activation and IL-1$\beta$ secretion could be reversed upon treatment of patients with anakinra [239]. These data suggest the potential therapeutic benefits of anakinra in decreasing inflammation associated with diverse diseases.

Another example is canakinumab (ILARIS®), a monoclonal antibody that targets IL-1β, a cytokine produced by inflammasomes [240]. The antibody binds to IL-1β, inhibiting its binding to the IL-1 receptor on cells and neutralizing its effects. Canakinumab has been approved for use in numerous inflammatory diseases, including RA and cryopyrin-associated periodic syndromes. The drug has also been approved for the treatment of gouty arthritis, a type of arthritis caused by the accumulation of uric acid crystals in joints, which can trigger the NLRP3 inflammasome [218]. In a clinical study, canakinumab was effective in reducing the number of gout flares in patients with a history of gout [219].

Glyburide (Glibenclamide) is an FDA-approved ATP-sensitive K+ channel inhibitor used for T2D in the United States [241] and has been recognized as an inhibitor of the NLRP3 inflammasome. Specifically, glyburide blocks the action of the ATP-sensitive potassium (KATP) channel, which regulates potassium flux in cells. Lamkanfi et al. conducted a study revealing that glyburide can block Cryopyrin activation and inhibit the secretion of IL-1β in response to microbial ligands, DAMPs, and crystals [220]. These data suggest the potential of glyburide as a therapeutic option for conditions such as gouty arthritis, silicosis, and AD, where the excessive production of IL-1β through Cryopyrin-dependent pathways is believed to significantly contribute to the underlying pathology [221,222].

Another inhibitor of the NLRP3 inflammasome is β-hydroxybutyrate, a ketone body produced during the breakdown of fatty acids [242]. β-hydroxybutyrate prevents inflammasome activation by triggering autophagy, in which cells degrade and recycle damaged or unwanted proteins and organelles [243]. Autophagy regulates the activation of the NLRP3 inflammasome by stimulating the degradation of its components. Therefore, β-hydroxybutyrate is a natural inhibitor of the NLRP3 inflammasome that promotes autophagy. In addition to inhibiting inflammasome activation, β-hydroxybutyrate has been found to have numerous other favorable effects on cellular function. For example, it is a potent energy source for the brain, and its production is boosted during fasting or calorie restriction [244]. These properties have led to an interest in β-hydroxybutyrate as a potential therapy for neurological disorders such as AD and Parkinson's disease. β-hydroxybutyrate has also been found to have anti-inflammatory effects in diverse contexts. In addition to its effect on the NLRP3 inflammasome, it has been shown to hinder the production of proinflammatory cytokines such as IL-1β and tumor necrosis factor-alpha (TNF-α) [245]. It may also regulate the activity of immune cells, including T cells and macrophages, although the precise mechanisms underlying these effects are not yet fully understood.

## 6. Conclusions and Future Perspectives

Several studies strongly propose that inflammasome complexes play an active role in the development of MS, AD, Parkinson's disease, atherosclerosis, and T2D. Moreover, inflammasomes are involved in other inflammatory conditions like rheumatoid arthritis, inflammatory bowel disease, gout, psoriasis, and systemic lupus erythematosus. Modulating inflammasome activation or targeting specific components of the pathway is an area of current research for potential therapeutic interventions in these settings. Although several components of the inflammasome complex have been identified, NLRP3 is a well-characterized inflammasome complex in multiple inflammatory conditions. As a result, numerous compounds have been investigated and recognized as potential inhibitors or modulators of NLRP3 inflammasome for treating inflammatory diseases. MCC950, ketotifen, fenamates, and tranilast have shown therapeutic efficiency against various inflammatory diseases, including MS, T2D, atherosclerosis, AD, gout, diabetic nephropathy, and diabetic encephalopathy. While these compounds have shown promise in preclinical studies and some clinical trials, more research is needed to evaluate their efficacy, safety, and specific mechanisms of action in targeting the NLRP3 inflammasome. Other compounds and natural products are also being investigated as potential NLRP3 inhibitors.

**Author Contributions:** Conceptualization, M.S.I.; writing—original draft preparation, S.S., T.D.V., T.A., E.K., L.M., A.K.M.M.M., M.M.A. and M.S.I.; writing—review and editing, S.S., T.D.V., T.A., E.K., L.M., A.K.M.M.M., M.M.A. and M.S.I.; supervision, A.K.M.M.M., M.M.A. and M.S.I. All authors have read and agreed to the published version of the manuscript.

**Funding:** This research received no external funding.

**Conflicts of Interest:** The authors declare no conflict of interest.

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
