# Peer review of "Exploring Inflammasome Complex as a Therapeutic Approach in Inflammatory Diseases"

_futurepharmacol, doi:10.3390/futurepharmacol3040048_

Round 1

Reviewer 1 Report

Comments and Suggestions for Authors

This is an excellent review article about a complex and very timely subject- inflammation. The authors cover many CNS and non-CNS/peripheral diseases affecting humans and demonstrate how various components of the immune response (esp activation of the NLRP3 inflammasome) are involved in pathogenesis. In addition, the authors provide an up-to-date summary in Table 1 of therapeutic trials of anti-inflammation strategies.

I feel that the article is of an appropriate length and is well structured and referenced. English language has a very few minor problems that are easily corrected with careful editing. This article belongs in the library of all investigators interested in CNS neurodegeneration (multiple sclerosis, AD, PD), T2DM/obesity and known inflammatory conditions (RA, IBD, etc). This article will no doubt stimulate additional laboratory and clinical investigations into inflammation, as well as hopefully an expansion of etiological thinking to include effects on inflammation cascades.

I have only one suggestion for the authors: please add a discussion about tissue selectivity of inflammasome activation. As a result of the efforts of many, it is now clear that inflammation is activated in tissues involved (CNS or peripheral) in various conditions, but it is unclear why inflammation is activated and/or why there is any tissue selectivity. I realize that much is not known in this area, but the authors will do readers a favor by discussing this void of knowledge. 

A minor point: llnes 553-556 may harbor a non-sequitur. In the preceding sentences the authors refer to studies showing that ASC deficiency reduces inflammation responses and increases resistance to metabolic deficiency. Then, in lines 553-556 the authors suggest that ASC deficiency itself can cause metabolic derangement. Which is correct?

However, this is a truly minor issue with clarity that is easily corrected. Overall, I am very enthusiastic about this paper and will be happy to review any changes the authors choose to make.

Author Response

Response: Manuscript ID: futurepharmacol-2498693

Dear Editor,

As a first step, the authors would like to express their appreciation to the three reviewers for their helpful suggestions regarding the manuscript entitled: “Targeting inflammasome complex in inflammatory diseases for possible therapeutics.”

We have addressed each of the reviewers’ comments point-by-point in the comments below. In the revised manuscript, altered text is shown by “red highlighted”. The authors believe the suggestions of the reviewers have greatly improved the manuscript.

We hope these changes will be satisfactory and look forward to any further suggestions.

Best wishes,

Md Soriful Islam,

for the authors

Reviewer 1

Response to Reviewer 1 Comment:

Comments and Suggestions for Authors

I have only one suggestion for the authors: please add a discussion about tissue selectivity of inflammasome activation.

Author’s Response: Thank you for the comment. The section has been added.

[Multiple studies have demonstrated how inflammation is activated in specific types of tissues under different conditions [11, 12]. One study found that increased IL-6 levels in pregnant mice result in cell-specific effects, specifically, immediate changes in fetal cells and long-term effects on downstream epithelial cell function [13]. However, in general, the mechanism for tissue selectivity of inflammasome activation is poorly understood due to a dearth of limited, direct studies. Understanding the tissue specificity of inflammasome activation is critical for developing novel targeted therapies.]

In the preceding sentences the authors refer to studies showing that ASC deficiency reduces inflammation responses and increases resistance to metabolic deficiency. Then, in lines 553-556 the authors suggest that ASC deficiency itself can cause metabolic derangement. Which is correct?

Author’s Response: Thank you for pointing this out. The latter comment has been deleted. In its place, we have included more information.

[“The ASC adaptor protein is an essential component of the inflammasome complex and acts as a bridge between NLRP3 and caspase-1 [176]. However, ASC-induced specks are not a prerequisite for inflammation activation but do maximize IL-1β processing [178].”]

Reviewer 2 Report

Comments and Suggestions for Authors

This manuscript concerns a currently fashionable topic but leaves the reader with a number of perplexities that should be corrected adequately to reach a high scientific standard: 

1. The title is misleading as it does not corresponds to the content of the paper, which by itself leaves out a lot of inflammatory diseases

2. The abstract is also somewhat misleading as the manuscript is not a comprehensive review of the inflammasomes, but considers only a few of them, actually mostly the NLRP3 one.

3. The general description of the inflammasomes' activation is rather synthetic (not all inflammasomes known are mentioned or even considered). What is sorely missing is some personal or original comments from the authors on this topic as well as on the few diseases considered. In most places it seems a listing of the literature published on each topic considered--information that can be easily found in other recently published reviews on the topic.

4. Figure 1 is quite similar to pictures that can be easily found in Google Images by typing "Inflammasomes". Conversely, noncanonical inflammasomes are not shown. Also, on the right side the upshots of inflammasomes' activity are missing. Moreover, the caption is too meager. 

5. Figure 5 is also misleading as NLRP3 is not the sole inflammasome importantly involved in Alzheimer's disease. Recent reviews in the Literature have highlighted this fact very clearly. Moreover, IL-1beta and IL-18 are among the products of inflammasomes' activity and not inflammasomes by themselves. 

6. Only in some places of the text a clear distinction is made between experimental data from relevant animal models of a disease and clinical data from the same disease-affected patients. No mention is made of the fact that animal models often do not completely mirror the human disease, which explains why only rarely drugs can be translated from animals to humans.

7. In Paragraph 5, the pharmacological compounds are listed iin a rather casual way with repetitions (e.g. MCC950) and mistakes (in Glyburide K+ efflux does not "block the formation of the NLRP3 inflammasome" but is necessary for NLRP3 ACTIVATION [see also Figure 1]).

8. Table 1 has a lot of unnecessary repetitions and inexplicably omits compounds like NSAIDs, TZDs, GLP-1, gamma-T3, and rosuvastatin that are mentioned in the text.

9. Several points in the text should be changed or emendated, e.g.:

9a.  Line 89: pyroptosis does not "clear infected cells" only;

9b. Lines 96-105. These examples could be more useful if moved to the specific paragraphs concerning AIM2 and NLRC4;

9c. Lines 341-346: the text does not strictly pertains to the AD topic and should be moved elsewhere;

9d. Lines 431-434: is AD or PD the real topic?

9e. Lines 503-504: the singular is in order as only one inflammasome has been mentioned in all this paragraph;

9f. Line 536.  "melanocortin receptor 4 SNPs" is an unclear mention;

9g. Line 644: "those": are they animals or humans?

10. The conclusions and future perspectives are weak and obvious: they do not report anything original and previously unpublished.

Comments on the Quality of English Language

The text has numerous typos, particularly missing spaces between words and reference numbers

Author Response

Response: Manuscript ID: futurepharmacol-2498693

Dear Editor,

As a first step, the authors would like to express their appreciation to the three reviewers for their helpful suggestions regarding the manuscript entitled: “Targeting inflammasome complex in inflammatory diseases for possible therapeutics.”

We have addressed each of the reviewers’ comments point-by-point in the comments below. In the revised manuscript, altered text is shown by “red highlighted”. The authors believe the suggestions of the reviewers have greatly improved the manuscript.

We hope these changes will be satisfactory and look forward to any further suggestions.

Best wishes,

Md Soriful Islam,

for the authors

Reviewer 2

Response to Reviewer 2 Comments:

  1. The title is misleading as it does not correspond to the content of the paper, which by itself leaves out a lot of inflammatory diseases

Author’s Response: Thank you for the comment. The title has been changed to better reflect the contents of the paper.       

[Working Title: “Exploring Inflammasome Complex as a Therapeutic Approach in Inflammatory Diseases”]

  1. The abstract is also somewhat misleading as the manuscript is not a comprehensive review of the inflammasomes, but considers only a few of them, actually mostly the NLRP3 one.

Author’s Response: Thank you for the comment. The abstract is now in the manuscript as follows:

[“Inflammasomes, a group of multi-protein complexes, are essential in regulating inflammation and immune responses. Several inflammasomes, including NLRP1, NLRP3, NLRP6, NLRP7, NLRP12, IFI16, NLRC4, AIM2, and pyrin, have been studied in various inflammatory diseases. Activating inflammasomes leads to the processing and production of pro-inflammatory cytokines, such as interleukin (IL)-1β and IL-18. The NLRP3 inflammasome is the most extensively studied and well-characterized. Consequently, targeting inflammasomes (particularly NLRP3) with several compounds, including small molecule inhibitors and natural compounds, has been studied as a potential therapeutic strategy. This review provides a comprehensive overview of different inflammasomes and their roles in six inflammatory diseases, including multiple sclerosis, Alzheimer’s disease, Parkinson’s disease, atherosclerosis, type 2 diabetes, and obesity. We also discussed different strategies that target inflammasomes to develop effective therapeutics.“]

  1. The general description of the inflammasomes' activation is rather synthetic (not all inflammasomes known are mentioned or even considered). What is sorely missing is some personal or original comments from the authors on this topic as well as on the few diseases considered. In most places it seems a listing of the literature published on each topic considered--information that can be easily found in other recently published reviews on the topic.

Author’s Response: Thanks for this suggestion. We acknowledge that inflammasome complexes are numerous, with varying levels of characterization. Thus, we have revised the manuscript to include additional information about inflammasomes, explicitly addressing the activation and role of NLRP6, NLRP7, NLRP12, and IFI16. However, we sought to present a balanced representation of the inflammasomes studied extensively in the investigated diseases, even though all are not listed.

Moreover, we want to clarify that the primary aim of our article, as outlined in the Methods section, is to provide a comprehensive overview of the existing literature concerning the significant role of inflammasomes in the most diffuse inflammatory diseases and the recent developments in therapeutic approaches. We acknowledge that numerous reviews in the literature discuss individual aspects of inflammasome activation or focus on specific diseases or particular therapeutic compounds. However, our manuscript is distinctive in its effort to synthesize and present a broad perspective by encompassing almost the whole spectrum of inflammasomes and the most diffuse associated diseases in a single cohesive narrative.

Our intent is not merely to provide "a listing" of previously published studies but a comprehensive account of the current knowledge on this complex subject. We believe that our article fills a critical gap in the existing literature by offering readers a consolidated and up-to-date resource. We also understand the desire for personal or original comments, but our primary focus remains to supply a detailed and unbiased overview of the existing literature within this domain.

  1. Figure 1 is quite similar to pictures that can be easily found in Google Images by typing "Inflammasomes". Conversely, noncanonical inflammasomes are not shown. Also, on the right side the upshots of inflammasomes' activity are missing. Moreover, the caption is too meager. 

Author’s Response: Thanks for this comment. We enhanced the caption of Figure 1 to provide a more comprehensive and informative description. The primary objective of Figure 1 is to provide a simple representation and a clear understanding of the general domain organization and activation process of the most well-known canonical inflammasomes. Specifically, the left side of the figure illustrates the domain organization of NLRP1, NLRP3, NLRC4, AIM2, Pyrin, ASC, and CASP1. We present a schematic view of the canonical activation process on the right side. It is essential to emphasize that this figure focuses only on canonical inflammasomes and is not intended to cover all inflammasome variations or noncanonical inflammasomes comprehensively.

We acknowledge the absence of noncanonical inflammasomes and the downstream consequences of inflammasome activation in the figure. However, this is a deliberate choice since the objective here is to provide an understanding of the structure of the key players under investigation and not of their downstream products.

  1. Figure 5 is also misleading as NLRP3 is not the sole inflammasome importantly involved in Alzheimer's disease. Recent reviews in the Literature have highlighted this fact very clearly. Moreover, IL-1beta and IL-18 are among the products of inflammasomes' activity and not inflammasomes by themselves. 

Author’s Response: Thanks for the suggestion. We added a caption to Figure 3 to explain, “The figure presents an overview of the major inflammasomes activated in the indicated diseases, along with the major products of inflammasome activity”. Moreover, we improved the manuscript introducing the role of NLRP1, AIM2 and NLRC4 in Alzheimer's disease.

  1. Only in some places of the text a clear distinction is made between experimental data from relevant animal models of a disease and clinical data from the same disease-affected patients. No mention is made of the fact that animal models often do not completely mirror the human disease, which explains why only rarely drugs can be translated from animals to humans.

Author’s Response: Thank you for the suggestion. We have added to the manuscript in line 773 as follows:

[“It is worth noting that animal models often do not mirror human diseases. More translational studies and clinical trials are required to determine whether these therapeutic strategies can be applied to humans.”]

  1. In Paragraph 5, the pharmacological compounds are listed in a rather casual way with repetitions (e.g. MCC950) and mistakes (in Glyburide K+efflux does not "block the formation of the NLRP3 inflammasome" but is necessary for NLRP3 ACTIVATION [see also Figure 1]).

Author’s Response: Thank you for the suggestion. We have made changes accordingly.

[“MCC950 (CP-456773, CRID3) is a potent small molecule inhibitor that mainly tar-gets the NLRP3 inflammasome.”

“A humanized antibody called IC100 (IgG4) has been specifically developed to tar-get the ASC component of inflammasomes.”

“Ketotifen (Zaditor®) is an antihistamine drug that is used to treat allergic rhinitis (hay fever) and allergic conjunctivitis.”

“Tranilast (N-[3’,4’-dimethoxycinnamoyl]-anthranilic acid), an anti-inflammatory agent mainly used in treating allergic disorders, has been assessed in preclinical and clin-ical studies for diverse inflammatory conditions.”

“For instance, anakinra (KINERET®), a recombinant form of the naturally occurring IL-1 receptor antagonist, binds to the same receptor as IL-1β but does not activate it.”

“Another example is canakinumab (ILARIS®), a monoclonal antibody that targets IL-1β, a cytokine produced by the inflammasome.”

“Glyburide (Glibenclamide) is an FDA-approved ATP-sensitive K+ channel inhibi-tor used for T2D in the United States.]

  1. Table 1 has a lot of unnecessary repetitions and inexplicably omits compounds like NSAIDs, TZDs, GLP-1, gamma-T3, and rosuvastatin that are mentioned in the text. Author’s Response: Thank you for pointing this out. We have included these compounds in the table.

  1. Several points in the text should be changed or emendated, e.g.:

Author’s Response: Thanks for this suggestion and for the efforts in reviewing our work. We have revised the manuscript to address text problems and inconsistencies.

  1. The conclusions and future perspectives are weak and obvious: they do not report anything original and previously unpublished.

Author’s Response: We appreciate this feedback and tried to enhance the clarity and relevance of our conclusions while remaining in line with the conventions of a narrative review that usually does not report unpublished or original data but provides a comprehensive overview of the current state of knowledge in the field. Our study intends to gather, organize, and evaluate the existing body of literature on the subject matter; therefore, while the manuscript's conclusions and future perspectives may appear to be "weak" or "obvious," they are based on a synthesis of previously published research summarizing the consensus and highlighting gaps in the current literature.

Reviewer 3 Report

Comments and Suggestions for Authors

This is a good summary for inflammasome.  Can the authors add some comments on NLRP6 inflammasome?

Author Response

Response: Manuscript ID: futurepharmacol-2498693

Dear Editor,

As a first step, the authors would like to express their appreciation to the three reviewers for their helpful suggestions regarding the manuscript entitled: “Targeting inflammasome complex in inflammatory diseases for possible therapeutics.”

We have addressed each of the reviewers’ comments point-by-point in the comments below. In the revised manuscript, altered text is shown by “red highlighted”. The authors believe the suggestions of the reviewers have greatly improved the manuscript.

We hope these changes will be satisfactory and look forward to any further suggestions.

Best wishes,

Md Soriful Islam,

for the authors

Reviewer 3

Response to Reviewer 3 Comment:

Comments and Suggestions for Authors

Reviewer 3: This is a good summary for inflammasome.  Can the authors add some comments on NLRP6 inflammasome?

Author’s Response: Thanks for this suggestion. We revised the manuscript to include evidence regarding NLRP6, NLRP7, NLRP12, and IFI16 inflammasomes since this would enhance the comprehensiveness of our summary.